# Data-Dependence of Plateau Phenomenon in Learning with Neural Network — Statistical Mechanical Analysis

**Yuki Yoshida**      **Masato Okada**

Department of Complexity Science and Engineering, Graduate School of Frontier Sciences,
The University of Tokyo
5-1-5 Kashiwanoha, Kashiwa, Chiba 277-8561, Japan
{yoshida@mns, okada@edu}.k.u-tokyo.ac.jp

## Abstract

The plateau phenomenon, wherein the loss value stops decreasing during the process of learning, has been reported by various researchers. The phenomenon is actively inspected in the 1990s and found to be due to the fundamental hierarchical structure of neural network models. Then the phenomenon has been thought as inevitable. However, the phenomenon seldom occurs in the context of recent deep learning. There is a gap between theory and reality. In this paper, using statistical mechanical formulation, we clarified the relationship between the plateau phenomenon and the statistical property of the data learned. It is shown that the data whose covariance has small and dispersed eigenvalues tend to make the plateau phenomenon inconspicuous.

## 1   Introduction

### 1.1   Plateau Phenomenon

Deep learning, and neural network as its essential component, has come to be applied to various fields. However, these still remain unclear in various points theoretically. The plateau phenomenon is one of them. In the learning process of neural networks, their weight parameters are updated iteratively so that the loss decreases. However, in some settings the loss does not decrease simply, but its decreasing speed slows down significantly partway through learning, and then it speeds up again after a long period of time. This is called as "plateau phenomenon". Since 1990s, this phenomena have been reported to occur in various practical learning situations (see Figure 1 (a) and Park et al. [2000], Fukumizu and Amari [2000]) . As a fundamental cause of this phenomenon, it has been pointed out by a number of researchers that the intrinsic symmetry of neural network models brings singularity to the metric in the parameter space which then gives rise to special attractors whose regions of attraction have nonzero measure, called as Milnor attractor (defined by Milnor [1985]; see also Figure 5 in Fukumizu and Amari [2000] for a schematic diagram of the attractor).

### 1.2   Who moved the plateau phenomenon?

However, the plateau phenomenon seldom occurs in recent practical use of neural networks (see Figure 1 (b) for example).

In this research, we rethink the plateau phenomenon, and discuss which situations are likely to cause the phenomenon. First we introduce the student-teacher model of two-layered networks as an ideal system. Next, we reduce the learning dynamics of the student-teacher model to a small-dimensional order parameter system by using statistical mechanical formulation, under the assumption that the

input dimension is sufficiently large. Through analyzing the order parameter system, we can discuss how the macroscopic learning dynamics depends on the statistics of input data. Our main contribution is the following:

- Under the statistical mechanical formulation of learning in the two-layered perceptron, we showed that macroscopic equations can be derived even when the statistical properties of the input are generalized. In other words, we extended the result of Saad and Solla [1995] and Riegler and Biehl [1995].

- By analyzing the macroscopic system we derived, we showed that the dynamics of learning depends only on the eigenvalue distribution of the covariance matrix of the input data.

- We clarified the relationship between the input data statistics and plateau phenomenon. In particular, it is shown that the data whose covariance matrix has small and disparsed eigenvalues tend to make the phenomenon inconspicuous, by numerically analyzing the macroscopic system.

## 1.3    Related works

The statistical mechanical approach used in this research is firstly developed by Saad and Solla [1995]. The method reduces high-dimensional learning dynamics of nonlinear neural networks to low-dimensional system of order parameters. They derived the macroscopic behavior of learning dynamics in two-layered soft-committee machine and by analyzing it they point out the existence of plateau phenomenon. Nowadays the statistical mechanical method is applied to analyze recent techniques (Hara et al. [2016], Yoshida et al. [2017], Takagi et al. [2019] and Straat and Biehl [2019]), and generalization performance in over-parameterized setting (Goldt et al. [2019]) and environment with conceptual drift (Straat et al. [2018]). However, it is unknown that how the property of input dataset itself can affect the learning dynamics, including plateaus.

Plateau phenomenon and singularity in loss landscape as its main cause have been studied by Fukumizu and Amari [2000], Wei et al. [2008], Cousseau et al. [2008] and Guo et al. [2018]. On the other hand, recent several works suggest that plateau and singularity can be mitigated in some settings. Orhan and Pitkow [2017] shows that skip connections eliminate the singularity. Another work by Yoshida et al. [2019] points out that output dimensionality affects the plateau phenomenon, in that multiple output units alleviate the plateau phenomenon. However, the number of output elements does not fully determine the presence or absence of plateaus, nor does the use of skip connections. The statistical property of data just can affect the learning dynamics dramatically; for example, see Figure 2 for learning curves with using different datasets and same network architecture. We focus on what kind of statistical property of the data brings plateau phenomenon.

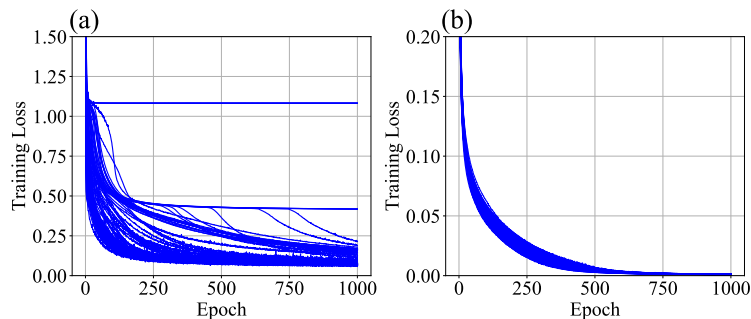

Figure 1:    (a) Training loss curves when two-layer perceptron with 4-4-3 units and ReLU activation learns IRIS dataset. (b) Training loss curve when two-layer perceptron with 784-20-10 units and ReLU activation learns MNIST dataset. For both (a) and (b), results of 100 trials with random initialization are overlaid. Minibatch size of 10 and vanilla SGD (learning rate: 0.01) are used.

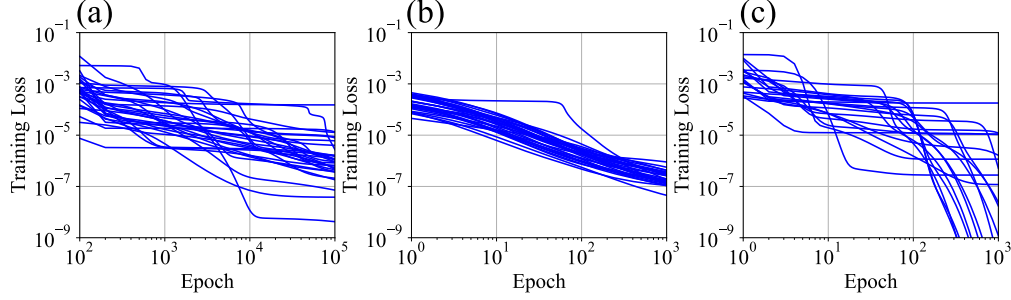

Figure 2: Loss curves yielded by student-teacher learning with two-layer perceptron which has 2 hidden units, 1 output unit and sigmoid activation, and with (a) IRIS dataset, (b) MNIST dataset, (c) a dataset in $\mathbb{R}^{60000 \times 784}$ drawn from standard normal distribution, as input distribution $p(\boldsymbol{\xi})$. In every subfigure, results for 20 trials with random initialization are overlaid. Vanilla SGD (learning rate: (a)(b) 0.005, (c) 0.001) and minibatch size of 1 are used for all three settings.

## 2 Formulation

### 2.1 Student-Teacher Model

We consider a two-layer perceptron which has $N$ input units, $K$ hidden units and 1 output unit. We denote the input to the network by $\boldsymbol{\xi} \in \mathbb{R}^N$. Then the output can be written as $s = \sum_{i=1}^{K} w_i g(\boldsymbol{J}_i \cdot \boldsymbol{\xi}) \in \mathbb{R}$, where $g$ is an activation function.

We consider the situation that the network learns data generated by another network, called "teacher network", which has fixed weights. Specifically, we consider two-layer perceptron that outputs $t = \sum_{n=1}^{M} v_n g(\boldsymbol{B}_n \cdot \boldsymbol{\xi}) \in \mathbb{R}$ for input $\boldsymbol{\xi}$ as the teacher network. The generated data $(\boldsymbol{\xi}, t)$ is then fed to the student network stated above and learned by it in the on-line manner (see Figure 3). We assume that the input $\boldsymbol{\xi}$ is drawn from some distribution $p(\boldsymbol{\xi})$ every time independently. We adopt vanilla stochastic gradient descent (SGD) algorithm for learning. We assume the squared loss function $\varepsilon = \frac{1}{2}(s - t)^2$, which is most commonly used for regression.

### 2.2 Statistical Mechanical Formulation

In order to capture the learning dynamics of nonlinear neural networks described in the previous subsection macroscopically, we introduce the statistical mechanical formulation in this subsection.

Let $x_i := \boldsymbol{J}_i \cdot \boldsymbol{\xi}$ ($1 \leq i \leq K$) and $y_n := \boldsymbol{B}_n \cdot \boldsymbol{\xi}$ ($1 \leq n \leq M$). Then

$$(x_1, \ldots, x_K, y_1, \ldots, y_M) \sim \mathcal{N}\left(0, [\boldsymbol{J}_1, \ldots, \boldsymbol{J}_K, \boldsymbol{B}_1, \ldots, \boldsymbol{B}_M]^T \Sigma [\boldsymbol{J}_1, \ldots, \boldsymbol{J}_K, \boldsymbol{B}_1, \ldots, \boldsymbol{B}_M]\right)$$

holds with $N \to \infty$ by generalized central limit theorem, provided that the input distribution $p(\boldsymbol{\xi})$ has zero mean and finite covariance matrix $\Sigma$.

Next, let us introduce order parameters as following: $Q_{ij} := \boldsymbol{J}_i^T \Sigma \boldsymbol{J}_j = \langle x_i x_j \rangle$, $R_{in} := \boldsymbol{J}_i^T \Sigma \boldsymbol{B}_n = \langle x_i y_n \rangle$, $T_{nm} := \boldsymbol{B}_n^T \Sigma \boldsymbol{B}_m = \langle y_n y_m \rangle$ and $D_{ij} := w_i w_j$, $E_{in} := w_i v_n$, $F_{nm} := v_n v_m$. Then

$$(x_1, \ldots, x_K, y_1, \ldots, y_M) \sim \mathcal{N}\left(\boldsymbol{0}, \begin{pmatrix} Q & R \\ R^T & T \end{pmatrix}\right).$$

The parameters $Q_{ij}$, $R_{in}$, $T_{nm}$, $D_{ij}$, $E_{in}$, and $F_{nm}$ introduced above capture the state of the system macroscopically; therefore they are called as "order parameters." The first three represent the state of the first layers of the two networks (student and teacher), and the latter three represent their second layers' state. $Q$ describes the statistics of the student's first layer and $T$ represents that of the teacher's first layer. $R$ is related to similarity between the student and teacher's first layer. $D, E, F$ is the second layers' counterpart of $Q, R, T$. The values of $Q_{ij}$, $R_{in}$, $D_{ij}$, and $E_{in}$ change during learning; their dynamics are what to be determined, from the dynamics of microscopic variables, i.e. connection weights. In contrast, $T_{nm}$ and $F_{nm}$ are constant during learning.

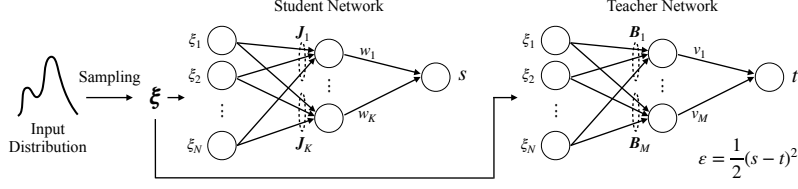

Figure 3: Overview of student-teacher model formulation.

### 2.2.1 Higher-order order parameters

The important difference between our situation and that of Saad and Solla [1995] is the covariance matrix $\Sigma$ of the input $\boldsymbol{\xi}$ is not necessarily equal to identity. This makes the matter complicated, since higher-order terms $\Sigma^e$ ($e = 1, 2, \ldots$) appear inevitably in the learning dynamics of order parameters. In order to deal with these, here we define some higher-order version of order parameters.

Let us define higher-order order parameters $Q_{ij}^{(e)}$, $R_{in}^{(e)}$ and $T_{nm}^{(e)}$ for $e = 0, 1, 2, \ldots$, as $Q_{ij}^{(e)} := \boldsymbol{J}_i^T \Sigma^e \boldsymbol{J}_j$, $\quad R_{in}^{(e)} := \boldsymbol{J}_i^T \Sigma^e \boldsymbol{B}_n$, $\quad$ and $\quad T_{nm}^{(e)} := \boldsymbol{B}_n^T \Sigma^e \boldsymbol{B}_m$. Note that they are identical to $Q_{ij}$, $R_{in}$ and $T_{nm}$ in the case of $e = 1$. Also we define higher-order version of $x_i$ and $y_n$, namely $x_i^{(e)}$ and $y_n^{(e)}$, as $x_i^{(e)} := \boldsymbol{\xi}^T \Sigma^e \boldsymbol{J}_i, y_n^{(e)} := \boldsymbol{\xi}^T \Sigma^e \boldsymbol{B}_n$. Note that $x_i^{(0)} = x_i$ and $y_n^{(0)} = y_n$.

## 3 Derivation of dynamics of order parameters

At each iteration of on-line learning, weights of the student network $\boldsymbol{J}_i$ and $w_i$ are updated with

$$\Delta \boldsymbol{J}_i = -\frac{\eta}{N}\frac{d\varepsilon}{d\boldsymbol{J}_i} = \frac{\eta}{N}[(\boldsymbol{t}-\boldsymbol{s}) \cdot \boldsymbol{w}_i]g'(x_i)\boldsymbol{\xi} = \frac{\eta}{N}\left[\left(\sum_{n=1}^{M} \boldsymbol{v}_n g(y_n) - \sum_{j=1}^{K} \boldsymbol{w}_j g(x_j)\right) \cdot \boldsymbol{w}_i\right] g'(x_i)\boldsymbol{\xi},$$

$$\Delta \boldsymbol{w}_i = -\frac{\eta}{N}\frac{d\varepsilon}{d\boldsymbol{w}_i} = \frac{\eta}{N}g(x_i)(\boldsymbol{t}-\boldsymbol{s}) = \frac{\eta}{N}g(x_i)\left(\sum_{n=1}^{M} \boldsymbol{v}_n g(y_n) - \sum_{j=1}^{K} \boldsymbol{w}_j g(x_j)\right),$$

(1)

in which we set the learning rate as $\eta/N$, so that our macroscopic system is $N$-independent.

Then, the order parameters $Q_{ij}^{(e)}$ and $R_{in}^{(e)}$ ($e = 0, 1, 2, \ldots$) are updated with

$$\Delta Q_{ij}^{(e)} = (\boldsymbol{J}_i + \Delta \boldsymbol{J}_i)^T \Sigma^e (\boldsymbol{J}_j + \Delta \boldsymbol{J}_j) - \boldsymbol{J}_i^T \Sigma^e \boldsymbol{J}_j = \boldsymbol{J}_i^T \Sigma^e \Delta \boldsymbol{J}_j + \boldsymbol{J}_j^T \Sigma^e \Delta \boldsymbol{J}_i + \Delta \boldsymbol{J}_i^T \Sigma^e \Delta \boldsymbol{J}_j$$

$$= \frac{\eta}{N}\left[\sum_{p=1}^{M} E_{ip}g'(x_i)x_j^{(e)}g(y_p) - \sum_{p=1}^{K} D_{ip}g'(x_i)x_j^{(e)}g(x_p)\right.$$

$$\left. + \sum_{p=1}^{M} E_{jp}g'(x_j)x_i^{(e)}g(y_p) - \sum_{p=1}^{K} D_{jp}g'(x_j)x_i^{(e)}g(x_p)\right]$$

$$+ \frac{\eta^2}{N^2}\boldsymbol{\xi}^T \Sigma^e \boldsymbol{\xi}\left[\sum_{p,q}^{K,K} D_{ip}D_{jq}g'(x_i)g'(x_j)g(x_p)g(x_q) + \sum_{p,q}^{M,M} E_{ip}E_{jq}g'(x_i)g'(x_j)g(y_p)g(y_q)\right.$$

$$\left. - \sum_{p,q}^{K,M} D_{ip}E_{jq}g'(x_i)g'(x_j)g(x_p)g(y_q) - \sum_{p,q}^{M,K} E_{ip}D_{jq}g'(x_i)g'(x_j)g(y_p)g(x_q)\right],$$

$$\Delta R_{in}^{(e)} = (\boldsymbol{J}_i + \Delta \boldsymbol{J}_i)^T \Sigma^e \boldsymbol{B}_n - \boldsymbol{J}_i^T \Sigma^e \boldsymbol{B}_n = \Delta \boldsymbol{J}_i^T \Sigma^e \boldsymbol{B}_n$$

$$= \frac{\eta}{N}\left[\sum_{p=1}^{M} E_{ip}g'(x_i)y_n^{(e)}g(y_p) - \sum_{p=1}^{K} D_{ip}g'(x_i)y_n^{(e)}g(x_p)\right].$$

(2)

Since

$$\boldsymbol{\xi}^T \Sigma^e \boldsymbol{\xi} \approx N\mu_{e+1} \qquad \text{where} \quad \mu_d := \frac{1}{N}\sum_{i=1}^{N}\lambda_i^d, \qquad \lambda_1, \ldots, \lambda_N : \text{eigenvalues of } \Sigma$$

and the right hand sides of the difference equations are $O(N^{-1})$, we can replace these difference equations with differential ones with $N \to \infty$, by taking the expectation over all input vectors $\boldsymbol{\xi}$:

$$
\begin{aligned}
\frac{dQ_{ij}^{(e)}}{d\tilde{\alpha}} = {} & \eta\left[\sum_{p=1}^{M} E_{ip}I_3(x_i, x_j^{(e)}, y_p) - \sum_{p=1}^{K} D_{ip}I_3(x_i, x_j^{(e)}, x_p)\right.\\
& \left. + \sum_{p=1}^{M} E_{jp}I_3(x_j, x_i^{(e)}, y_p) - \sum_{p=1}^{K} D_{jp}I_3(x_j, x_i^{(e)}, x_p)\right]\\
& + \eta^2\mu_{e+1}\left[\sum_{p,q}^{K,K} D_{ip}D_{jq}I_4(x_i, x_j, x_p, x_q) + \sum_{p,q}^{M,M} E_{ip}E_{jq}I_4(x_i, x_j, y_p, y_q)\right.\\
& \left. - \sum_{p,q}^{K,M} D_{ip}E_{jq}I_4(x_i, x_j, x_p, y_q) - \sum_{p,q}^{M,K} E_{ip}D_{jq}I_4(x_i, x_j, y_p, x_q)\right],
\end{aligned}
\tag{3}
$$

$$\frac{dR_{in}^{(e)}}{d\tilde{\alpha}} = \eta\left[\sum_{p=1}^{M} E_{ip}I_3(x_i, y_n^{(e)}, y_p) - \sum_{p=1}^{K} D_{ip}I_3(x_i, y_n^{(e)}, x_p)\right]$$

where $\quad I_3(z_1, z_2, z_3) := \langle g'(z_1)z_2 g(z_3)\rangle \quad$ and $\quad I_4(z_1, z_2, z_3, z_4) := \langle g'(z_1)g'(z_2)g(z_3)g(z_4)\rangle.$
$$\tag{4}$$

In these equations, $\tilde{\alpha} := \alpha/N$ represents time (normalized number of steps), and the brackets $\langle\cdot\rangle$ represent the expectation when the input $\boldsymbol{\xi}$ follows the input distribution $p(\boldsymbol{\xi})$.

The differential equations for $D$ and $E$ are obtained in a similar way:

$$
\begin{aligned}
\frac{dD_{ij}}{d\tilde{\alpha}} &= \eta\left[\sum_{p=1}^{M} E_{ip}I_2(x_j, y_p) - \sum_{p=1}^{K} D_{ip}I_2(x_j, x_p) + \sum_{p=1}^{M} E_{jp}I_2(x_i, y_p) - \sum_{p=1}^{K} D_{jp}I_2(x_i, x_p)\right],\\
\frac{dE_{in}}{d\tilde{\alpha}} &= \eta\left[\sum_{p=1}^{M} F_{pn}I_2(x_i, y_p) - \sum_{p=1}^{K} E_{pn}I_2(x_i, x_p)\right]
\end{aligned}
$$
$$\tag{5}$$

$$\text{where} \quad I_2(z_1, z_2) := \langle g(z_1)g(z_2)\rangle. \tag{6}$$

These differential equations (3) and (5) govern the macroscopic dynamics of learning. In addition, the generalization loss $\varepsilon_g$, the expectation of loss value $\varepsilon(\boldsymbol{\xi}) = \frac{1}{2}\|\boldsymbol{s} - \boldsymbol{t}\|^2$ over all input vectors $\boldsymbol{\xi}$, is represented as

$$\varepsilon_g = \langle \frac{1}{2}\|\boldsymbol{s} - \boldsymbol{t}\|^2\rangle = \frac{1}{2}\left[\sum_{p,q}^{M,M} F_{pq}I_2(y_p, y_q) + \sum_{p,q}^{K,K} D_{pq}I_2(x_p, x_q) - 2\sum_{p,q}^{K,M} E_{pq}I_2(x_p, y_q)\right].$$
$$\tag{7}$$

### 3.1 Expectation terms

Above we have determined the dynamics of order parameters as (3), (5) and (7). However they have expectation terms $I_2(z_1, z_2)$, $I_3(z_1, z_2^{(e)}, z_3)$ and $I_4(z_1, z_2, z_3, z_4)$, where $z$s are either $x_i$ or $y_n$. By studying what distribution $\boldsymbol{z}$ follows, we can show that these expectation terms are dependent only on 1-st and $(e+1)$-th order parameters, namely, $Q^{(1)}, R^{(1)}, T^{(1)}$ and $Q^{(e+1)}, R^{(e+1)}, T^{(e+1)}$; for example,

$$I_3(x_i, x_j^{(e)}, y_p) = \int dz_1 dz_2 dz_3 \, g'(z_1)z_2 g(z_3)\, \mathcal{N}\!\left(\boldsymbol{z}\,\Big|\,\boldsymbol{0}, \begin{pmatrix} Q_{ii}^{(1)} & Q_{ij}^{(e+1)} & R_{ip}^{(1)} \\ Q_{ij}^{(e+1)} & * & R_{jp}^{(e+1)} \\ R_{ip}^{(1)} & R_{jp}^{(e+1)} & T_{pp}^{(1)} \end{pmatrix}\right)$$

holds, where $*$ does not influence the value of this expression (see Supplementary Material A.1 for more detailed discussion). Thus, we see the 'speed' of $e$-th order parameters (i.e. (3) and (5)) only depends on 1-st and $(e+1)$-th order parameters, and the generalization error $\varepsilon_g$ (equation (7)) only depends on 1-st order parameters. Therefore, with denoting $(Q^{(e)}, R^{(e)}, T^{(e)})$ by $\Omega^{(e)}$ and $(D, E, F)$ by $\chi$, we can write

$$\frac{d}{d\tilde{\alpha}}\Omega^{(e)} = f^{(e)}(\Omega^{(1)}, \Omega^{(e+1)}, \chi), \qquad \frac{d}{d\tilde{\alpha}}\chi = g(\Omega^{(1)}, \chi), \quad \text{and} \quad \varepsilon_g = h(\Omega^{(1)}, \chi)$$

with appropriate functions $f^{(e)}$, $g$ and $h$. Additionally, a polynomial of $\Sigma$

$$P(\Sigma) := \prod_{i=1}^{d}(\Sigma - \lambda_i' I_N) = \sum_{e=0}^{d} c_e \Sigma^e \qquad \text{where} \quad \lambda_1', \dots, \lambda_d' \quad \text{are distinct eigenvalues of } \Sigma$$

equals to 0, thus we get

$$\Omega^{(d)} = -\sum_{e=0}^{d-1} c_e \Omega^{(e)}. \tag{8}$$

Using this relation, we can reduce $\Omega^{(d)}$ to expressions which contain only $\Omega^{(0)}, \dots, \Omega^{(d-1)}$, therefore we can get a closed differential equation system with $\Omega^{(0)}, \dots, \Omega^{(d-1)}$ and $\chi$.

In summary, our macroscopic system is closed with the following order parameters:

Order variables : $\quad Q_{ij}^{(0)}, Q_{ij}^{(1)}, \dots, Q_{ij}^{(d-1)}, \quad R_{in}^{(0)}, R_{in}^{(1)}, \dots, R_{in}^{(d-1)}, \quad D_{ij}, E_{in}$

Order constants : $\quad T_{nm}^{(0)}, T_{nm}^{(1)}, \dots, T_{nm}^{(d-1)}, \quad F_{nm}. \qquad$ ($d$: number of distinct eigenvalues of $\Sigma$)

The order variables are governed by (3) and (5). For the lengthy full expressions of our macroscopic system for specific cases, see Supplementary Material A.2.

### 3.2 Dependency on input data covariance $\Sigma$

The differential equation system we derived depends on $\Sigma$, through two ways; the coefficient $\mu_{e+1}$ of $O(\eta^2)$-term, and how ($d$)-th order parameters are expanded with lower order parameters (as (8)). Specifically, the system only depends on the eigenvalue distribution of $\Sigma$.

### 3.3 Evaluation of expectation terms for specific activation functions

Expectation terms $I_2$, $I_3$ and $I_4$ can be analytically determined for some activation functions $g$, including sigmoid-like $g(x) = \text{erf}(x/\sqrt{2})$ (see Saad and Solla [1995]) and $g(x) = \text{ReLU}(x)$ (see Yoshida et al. [2017]).

## 4 Analysis of numerical solutions of macroscopic differential equations

In this section, we analyze numerically the order parameter system, derived in the previous section[1]. We assume that the second layers' weights of the student and the teacher, namely $w_i$ and $v_n$, are fixed to 1 (i.e. we consider the learning of soft-committee machine), and that $K$ and $M$ are equal to 2, for simplicity. Here we think of sigmoid-like activation $g(x) = \text{erf}(x/\sqrt{2})$.

### 4.1 Consistency between macroscopic system and microscopic system

First of all, we confirmed the consistency between the macroscopic system we derived and the original microscopic system. That is, we computed the dynamics of the generalization loss $\varepsilon_g$ in two ways: (i) by updating weights of the network with SGD (1) iteratively, and (ii) by solving numerically the differential equations (5) which govern the order parameters, and we confirmed that they accord with each other very well (Figure 4). Note that we set the initial values of order parameters in (ii) as values corresponding to initial weights used in (i). For dependence of the learning trajectory on the initial condition, see Supplementary Material A.3.

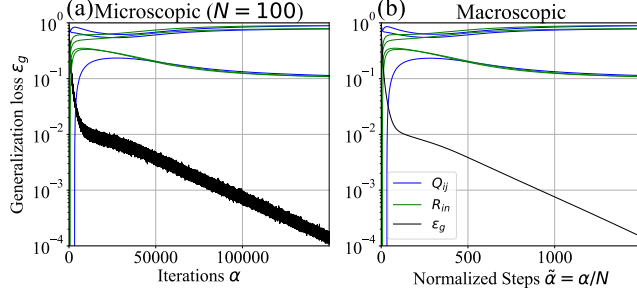

Figure 4: Example dynamics of generalization error $\varepsilon_g$ computed with (a) microscopic and (b) macroscopic system. Network size: $N$-2-1. Learning rate: $\eta = 0.1$. Eigenvalues of $\Sigma$: $\lambda_1 = 0.4$ with multiplicity $0.5N$, $\lambda_2 = 1.2$ with multiplicity $0.3N$, and $\lambda_3 = 1.6$ with multiplicity $0.2N$. Black lines: dynamics of $\varepsilon_g$. Blue lines: $Q_{11}, Q_{12}, Q_{22}$. Green lines: $R_{11}, R_{12}, R_{21}, R_{22}$.

## 4.2 Case of scalar input covariance $\Sigma = \sigma I_N$

As the simplest case, here we consider the case that the convariance matrix $\Sigma$ is proportional to unit matrix. In this case, $\Sigma$ has only one eigenvalue $\lambda = \mu_1$ of multiplicity $N$, then our order parameter system contains only parameters whose order is $0$ ($e = 0$). For various values of $\mu_1$, we solved numerically the differential equations of order parameters (5) and plotted the time evolution of generalization loss $\varepsilon_g$ (Figure 5(a)). From these plots, we quantified the lengths and heights of the plateaus as following: we regarded the system is plateauing if the decreasing speed of log-loss is smaller than half of its terminal converging speed, and we defined the height of the plateau as the median of loss values during plateauing. Quantified lengths and heights are plotted in Figure 5(b)(c). It indicates that the plateau length and height heavily depend on $\mu_1$, the input scale. Specifically, as $\mu_1$ decreases, the plateau rapidly becomes longer and lower. Though smaller input data lead to longer plateaus, it also becomes lower and then inconspicuous. This tendency is consistent with Figure 2(a)(b), since IRIS dataset has large $\mu_1$ ($\approx 15.9$) and MNIST has small $\mu_1$ ($\approx 0.112$). Considering this, the claim that the plateau phenomenon does not occur in learning of MNIST is controversy; this suggests the possibility that we are observing quite long and low plateaus.

Note that Figure 5(b) shows that the speed of growing of plateau length is larger than $O(1/\mu_1)$. This is contrast to the case of linear networks which have no activation; in that case, as $\mu_1$ decreases the speed of learning gets exactly $1/\mu_1$-times larger. In other words, this phenomenon is peculiar to nonlinear networks.

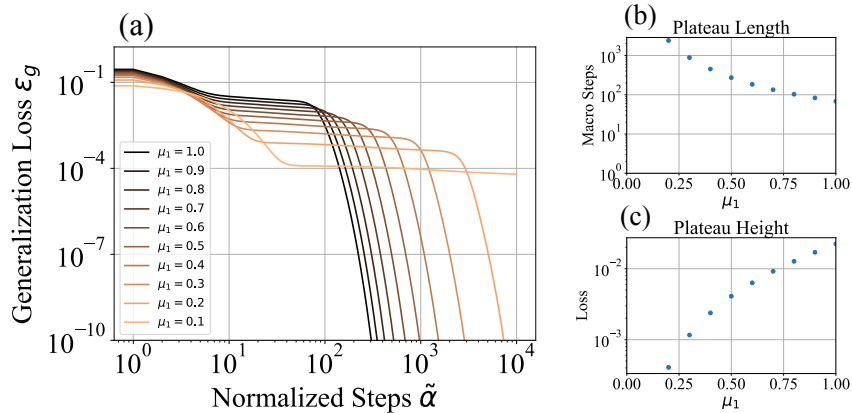

Figure 5: (a) Dynamics of generalization error $\varepsilon_g$ when input variance $\Sigma$ has only one eigenvalue $\lambda = \mu_1$ of multiplicity $N$. Plots with various values of $\mu_1$ are shown. (b) Plateau length and (b) plateau height, quantified from (a).

### 4.3 Case of different input covariance $\Sigma$ with fixed $\mu_1$

In the previous subsection we inspected the dependence of the learning dynamics on the first moment $\mu_1$ of the eigenvalues of the covariance matrix $\Sigma$. In this subsection, we explored the dependence of the dynamics on the higher moments of eigenvalues, under fixed first moment $\mu_1$.

In this subsection, we consider the case in which the input covariance matrix $\Sigma$ has two distinct nonzero eigenvalues, $\lambda_1 = \mu_1 - \Delta\lambda/2$ and $\lambda_2 = \mu_1 + \Delta\lambda/2$, of the same multiplicity $N/2$ (Figure 6). With changing the control parameter $\Delta\lambda$, we can get eigenvalue distributions with various values of second moment $\mu_2 = \langle \lambda_i^2 \rangle$.

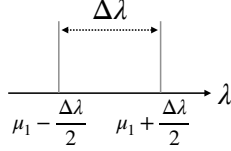

Figure 6: Eigenvalue distribution with fixed $\mu_1$ parameterized by $\Delta\lambda$, which yields various $\mu_2$.

Figure 7(a) shows learning curves with various $\mu_2$ while fixing $\mu_1$ to 1. From these curves, we quantified the lengths and heights of the plateaus, and plotted them in Figure 7(b)(c). These indicate that the length of the plateau shortens as $\mu_2$ becomes large. That is, the more the distribution of nonzero eigenvalues gets broaden, the more the plateau gets alleviated.

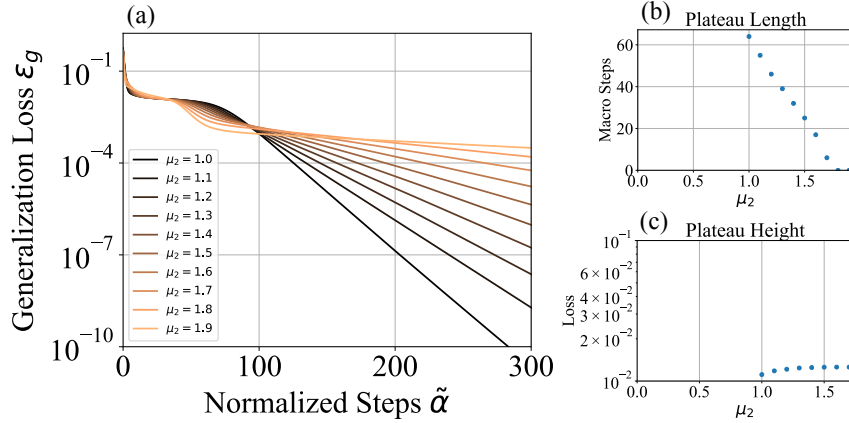

Figure 7: (a) Dynamics of generalization error $\varepsilon_g$ when input variance $\Sigma$ has two eigenvalues $\lambda_{1,2} = \mu_1 \pm \Delta\lambda/2$ of multiplicity $N/2$. Plots with various values of $\mu_2$ are shown. (b) Plateau length and (c) plateau height, quantified from (a).

## 5 Conclusion

Under the statistical mechanical formulation of learning in the two-layered perceptron, we showed that macroscopic equations can be derived even when the statistical properties of the input are generalized. We showed that the dynamics of learning depends only on the eigenvalue distribution of the covariance matrix of the input data. By numerically analyzing the macroscopic system, it is shown that the statistics of input data dramatically affect the plateau phenomenon.

Through this work, we explored the gap between theory and reality; though the plateau phenomenon is theoretically predicted to occur by the general symmetrical structure of neural networks, it is seldom observed in practice. However, more extensive researches are needed to fully understand the theory underlying the plateau phenomenon in practical cases.

## Acknowledgement

This work was supported by JSPS KAKENHI Grant-in-Aid for Scientific Research(A) (No. 18H04106).

## Footnotes

[1] We executed all computations on a standard PC.

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
