[Supplementary Material]

# Data-Dependence of Plateau Phenomenon in Learning with Neural Network — Statistical Mechanical Analysis (Supplementary Material)

**Yuki Yoshida**          **Masato Okada**

Department of Complexity Science and Engineering, Graduate School of Frontier Sciences,
The University of Tokyo
5-1-5 Kashiwanoha, Kashiwa, Chiba 277-8561, Japan
{yoshida@mns, okada@edu}.k.u-tokyo.ac.jp

## A.1  Properties of expectation term $I_2$, $I_3$ and $I_4$

The differential equations of learning dynamics (3) and (5) in the main text have expectation terms, $I_2(z_1, z_2)$, $I_3(z_1, z_2, z_3)$ and $I_4(z_1, z_2, z_3, z_4)$. Since their $z$s are either $x_i^{(e)} = \boldsymbol{\xi}^T \Sigma^e \boldsymbol{J}_i$ or $y_n^{(e)} = \boldsymbol{\xi}^T \Sigma^e \boldsymbol{B}_n$, any tuple $(z_1, z_2, \dots)$ follows multivariate normal distribution $\mathcal{N}(\boldsymbol{z}|0, \langle \boldsymbol{z} \cdot \boldsymbol{z}^T \rangle)$ when $N \to \infty$ by generalized central limit theorem, provided that the input $\boldsymbol{\xi}$ has zero mean and finite covariance. Thus the expectation terms only depend on the covariance matrix $\langle \boldsymbol{z} \cdot \boldsymbol{z}^T \rangle$, and their elements can be calculated as $\langle x_i^{(e)} x_j^{(f)} \rangle = Q_{ij}^{(e+f+1)}$, $\langle x_i^{(e)} y_n^{(f)} \rangle = R_{in}^{(e+f+1)}$ and $\langle y_n^{(e)} y_m^{(f)} \rangle = T_{nm}^{(e+f+1)}$. For example,

$$I_2(x_i, y_p) = \int dz_1 dz_2 \, g(z_1) g(z_2) \, \mathcal{N}\left(\boldsymbol{z}|\boldsymbol{0}, \begin{pmatrix} Q_{ii}^{(1)} & R_{ip}^{(1)} \\ & T_{pp}^{(1)} \end{pmatrix}\right),$$

$$I_3(x_i, x_j^{(e)}, y_p) = \int dz_1 dz_2 dz_3 \, g'(z_1) z_2 g(z_3) \, \mathcal{N}\left(\boldsymbol{z}|\boldsymbol{0}, \begin{pmatrix} Q_{ii}^{(1)} & Q_{ij}^{(e+1)} & R_{ip}^{(1)} \\ & Q_{jj}^{(2e+1)} & R_{jp}^{(e+1)} \\ & & T_{pp}^{(1)} \end{pmatrix}\right),$$

$$I_4(x_i, x_j, y_p, y_q) = \int dz_1 dz_2 dz_3 dz_4 \, g(z_1) g(z_2) g(z_3) g(z_4) \, \mathcal{N}\left(\boldsymbol{z}|\boldsymbol{0}, \begin{pmatrix} Q_{ii}^{(1)} & Q_{ij}^{(1)} & R_{ip}^{(1)} & R_{iq}^{(1)} \\ & Q_{jj}^{(1)} & R_{jp}^{(1)} & R_{jq}^{(1)} \\ & & T_{pp}^{(1)} & T_{pq}^{(1)} \\ & & & T_{qq}^{(1)} \end{pmatrix}\right).$$

Note that all the covariance matrix is symmetric. Their left-bottom sides are not shown for notational simplicity. Substituting these for $I$s shown in equations (3) and (5) in the main text, we see that the 'speed' of $e$-th order parameters can be dependent only on 1-st, $(e+1)$-th, and $(2e+1)$-th order parameters.

Here we prove the following proposition, in order to show that the 'speed' of $e$-th order parameters are not dependent on $(2e+1)$-th order parameters.

**Proposition.**   The expectation term $I_3(z_1, z_2, z_3) := \int dz_1 dz_2 dz_3 \, g'(z_1) z_2 g(z_3) \, \mathcal{N}(\boldsymbol{z}|\boldsymbol{0}, C)$ does not depend on $C_{22}$.

**Proof.**   Since $C$ is positive-semidefinite, we can write $C = VV^T$ for some squared matrix $V$. Thus, when $\boldsymbol{\xi} \sim \mathcal{N}(0, I_N)$, $A\boldsymbol{\xi} \sim \mathcal{N}(0, C)$ holds. Therefore, we can regard that $z_i (i = 1, 2, 3)$ is generated by $z_i = \boldsymbol{v}_i^T \boldsymbol{\xi}$ where $\boldsymbol{v}_i$ is $i$-th row vector of $V$ and $\boldsymbol{\xi}$ follows the standard normal distribution.

We can write $\boldsymbol{v}_2 = c_1\boldsymbol{v}_1 + c_3\boldsymbol{v}_3 + \boldsymbol{v}^\perp$ for some coefficient $c_1, c_3 \in \mathbb{R}$ and some vector $\boldsymbol{v}^\perp$ perpendicular to $\boldsymbol{v}_1$ and $\boldsymbol{v}_3$. Then $I_3$ is written as

$$I_3(z_1, z_2, z_3) = \langle g'(z_1)z_2 g(z_3)\rangle = c_1\langle g'(z_1)z_1 g(z_3)\rangle + c_3\langle g'(z_1)z_3 g(z_3)\rangle + \langle g'(z_1)\boldsymbol{v}^{\perp T}\boldsymbol{\xi}g(z_3)\rangle.$$

Since $\boldsymbol{\xi} \sim \mathcal{N}(0, I_N)$ and $\boldsymbol{v}^\perp \perp \boldsymbol{v}_1, \boldsymbol{v}_3$ hold, $(z_1, z_3)$ and $\boldsymbol{v}^{\perp T}\boldsymbol{\xi}$ is independent. Therefore the third term in the right hand side of the equation above is

$$\langle g'(z_1)\boldsymbol{v}^{\perp T}\boldsymbol{\xi}g(z_3)\rangle = \langle g'(z_1)g(z_3)\rangle\langle \boldsymbol{v}^{\perp T}\boldsymbol{\xi}\rangle = 0.$$

In addition, we can determine $c_1$ and $c_3$ by solving

$$C_{12} = \boldsymbol{v}_2^T\boldsymbol{v}_1 = (c_1\boldsymbol{v}_1^T + c_3\boldsymbol{v}_3^T + \boldsymbol{v}^{\perp T})\boldsymbol{v}_1 = c_1 C_{11} + c_3 C_{13} \quad \text{and}$$
$$C_{23} = \boldsymbol{v}_2^T\boldsymbol{v}_3 = (c_1\boldsymbol{v}_1^T + c_3\boldsymbol{v}_3^T + \boldsymbol{v}^{\perp T})\boldsymbol{v}_3 = c_1 C_{13} + c_3 C_{33}.$$

Together with these, we get

$$I_3(z_1, z_2, z_3) = \frac{(C_{12}C_{33} - C_{13}C_{23})\, I_3(z_1, z_1, z_3) + (C_{11}C_{23} - C_{12}C_{13})\, I_3(z_1, z_3, z_3)}{C_{11}C_{33} - C_{13}^2},$$

which shows that $I_3$ is independent to $C_{22}$. ∎

## A.2 Full expression of order parameter system

Here we describe the whole system of the order parameters, with specific eigenvalue distribution of $\Sigma$.

### A.2.1 Case with $\Sigma = \sigma I_N$

In this case, the order parameters are

$$\text{Order variables :} \qquad Q_{ij}^{(0)}, \qquad R_{in}^{(0)}, \qquad D_{ij}, E_{in}$$
$$\text{Order constants :} \qquad T_{nm}^{(0)}, \qquad F_{nm}.$$

Note that $Q_{ij}^{(1)}$ is identical to $Q_{ij}^{(0)}$. This is same for $R$ and $T$. The order parameter system is described as following, with omitting $^{(0)}$-s for notational simplicity:

$$\begin{aligned}
\frac{dQ_{ij}}{d\tilde{\alpha}} = \eta &\left[\sum_{p=1}^{M} E_{ip}I_3\begin{pmatrix} Q_{ii} & Q_{ij} & R_{ip} \\ & * & R_{jp} \\ & & T_{pp} \end{pmatrix} - \sum_{p=1}^{K} D_{ip}I_3\begin{pmatrix} Q_{ii} & Q_{ij} & Q_{ip} \\ & * & Q_{jp} \\ & & Q_{pp} \end{pmatrix}\right. \\
&\left. + \sum_{p=1}^{M} E_{jp}I_3\begin{pmatrix} Q_{jj} & Q_{ji} & R_{jp} \\ & * & R_{ip} \\ & & T_{pp} \end{pmatrix} - \sum_{p=1}^{K} D_{jp}I_3\begin{pmatrix} Q_{jj} & Q_{ji} & Q_{jp} \\ & * & Q_{ip} \\ & & Q_{pp} \end{pmatrix}\right] \\
+ \eta^2 &\left[\sum_{p,q}^{K,K} D_{ip}D_{jq}I_4\begin{pmatrix} Q_{ii} & Q_{ij} & Q_{ip} & Q_{iq} \\ & Q_{jj} & Q_{jp} & Q_{jq} \\ & & Q_{pp} & Q_{pq} \\ & & & Q_{qq} \end{pmatrix} + \sum_{p,q}^{M,M} E_{ip}E_{jq}I_4\begin{pmatrix} Q_{ii} & Q_{ij} & R_{ip} & R_{iq} \\ & Q_{jj} & R_{jp} & R_{jq} \\ & & T_{pp} & T_{pq} \\ & & & T_{qq} \end{pmatrix}\right. \\
&\left. - \sum_{p,q}^{K,M} D_{ip}E_{jq}I_4\begin{pmatrix} Q_{ii} & Q_{ij} & Q_{ip} & R_{iq} \\ & Q_{jj} & Q_{jp} & R_{jq} \\ & & Q_{pp} & R_{pq} \\ & & & T_{qq} \end{pmatrix} - \sum_{p,q}^{M,K} E_{ip}D_{jq}I_4\begin{pmatrix} Q_{ii} & Q_{ij} & R_{ip} & Q_{iq} \\ & Q_{jj} & R_{jp} & Q_{jq} \\ & & T_{pp} & R_{pq} \\ & & & Q_{qq} \end{pmatrix}\right], \\
\frac{dR_{in}}{d\tilde{\alpha}} = \eta &\left[\sum_{p=1}^{M} E_{ip}I_3\begin{pmatrix} Q_{ii} & R_{in} & R_{ip} \\ & * & T_{np} \\ & & T_{pp} \end{pmatrix} - \sum_{p=1}^{K} D_{ip}I_3\begin{pmatrix} Q_{ii} & R_{in} & Q_{ip} \\ & * & R_{pn} \\ & & Q_{pp} \end{pmatrix}\right]
\end{aligned}$$

$$\tag{1}$$

and

$$\frac{dD_{ij}}{d\tilde{\alpha}} = \eta \left[ \sum_{p=1}^{M} E_{ip} I_2 \left( \begin{array}{cc} Q_{jj} & R_{jp} \\ & T_{pp} \end{array} \right) - \sum_{p=1}^{K} D_{ip} I_2 \left( \begin{array}{cc} Q_{jj} & Q_{jp} \\ & Q_{pp} \end{array} \right) \right.$$

$$\left. + \sum_{p=1}^{M} E_{jp} I_2 \left( \begin{array}{cc} Q_{ii} & R_{ip} \\ & T_{pp} \end{array} \right) - \sum_{p=1}^{K} D_{jp} I_2 \left( \begin{array}{cc} Q_{ii} & Q_{ip} \\ & Q_{pp} \end{array} \right) \right], \qquad (2)$$

$$\frac{dE_{in}}{d\tilde{\alpha}} = \eta \left[ \sum_{p=1}^{M} F_{pn} I_2 \left( \begin{array}{cc} Q_{ii} & R_{ip} \\ & T_{pp} \end{array} \right) - \sum_{p=1}^{K} E_{pn} I_2 \left( \begin{array}{cc} Q_{jj} & R_{jp} \\ & T_{pp} \end{array} \right) \right]$$

,

where $\quad I_2(C) = \dfrac{2}{\pi} \arcsin \dfrac{C_{12}}{\sqrt{1+C_{11}}\sqrt{1+C_{22}}},$

$$I_3(C) = \frac{2}{\pi} \cdot \frac{1}{\sqrt{(1+C_{11})(1+C_{33}) - C_{13}^2}} \frac{C_{23}(1+C_{11}) - C_{12}C_{13}}{1+C_{11}},$$

$$I_4(C) = \frac{4}{\pi^2} \cdot \frac{1}{\sqrt{1+2C_{11}}} \arcsin \frac{(1+2C_{11})C_{23} - 2C_{12}C_{13}}{\sqrt{(1+2C_{11})(1+2C_{22}) - 2C_{12}^2}\sqrt{(1+2C_{11})(1+2C_{33}) - 2C_{13}^2}}$$

(3)

for $g(x) = \mathrm{erf}(x/\sqrt{2})$ activation, as **?** showed.

### A.2.2 Case with $\Sigma$ which has two distinct eigenvalues, $\lambda_1$ of multiplicity $r_1 N$ and $\lambda_2$ of multiplicity $r_2 N$

In this case, the order parameters are

$$\begin{aligned} \text{Order variables}: \quad & Q_{ij}^{(0)}, Q_{ij}^{(1)}, \quad R_{in}^{(0)}, R_{in}^{(1)}, \quad D_{ij}, E_{in} \\ \text{Order constants}: \quad & T_{nm}^{(0)}, T_{nm}^{(1)}, \quad F_{nm}. \end{aligned}$$

Since $\Sigma^2 - (\lambda_1 + \lambda_2)\Sigma + \lambda_1\lambda_2 I_N = 0$, the relation $Q_{ij}^{(2)} = (\lambda_1 + \lambda_2)Q_{ij}^{(1)} - \lambda_1\lambda_2 Q_{ij}^{(0)}$ holds. This is same for $R$ and $T$. Then the order parameter system is described as following:

$$
\begin{aligned}
\frac{dQ_{ij}^{(0)}}{d\tilde{\alpha}} = \eta &\left[ \sum_{p=1}^{M} E_{ip}I_3 \begin{pmatrix} Q_{ii}^{(1)} & Q_{ij}^{(1)} & R_{ip}^{(1)} \\ & * & R_{jp}^{(1)} \\ & & T_{pp}^{(1)} \end{pmatrix} - \sum_{p=1}^{K} D_{ip}I_3 \begin{pmatrix} Q_{ii}^{(1)} & Q_{ij}^{(1)} & Q_{ip}^{(1)} \\ & * & Q_{jp}^{(1)} \\ & & Q_{pp}^{(1)} \end{pmatrix} \right. \\
&+ \sum_{p=1}^{M} E_{jp}I_3 \begin{pmatrix} Q_{jj}^{(1)} & Q_{ji}^{(1)} & R_{jp}^{(1)} \\ & * & R_{ip}^{(1)} \\ & & T_{pp}^{(1)} \end{pmatrix} - \sum_{p=1}^{K} D_{jp}I_3 \begin{pmatrix} Q_{jj}^{(1)} & Q_{ji}^{(1)} & Q_{jp}^{(1)} \\ & * & Q_{ip}^{(1)} \\ & & Q_{pp}^{(1)} \end{pmatrix} \left. \right] \\
&+ \eta^2(r_1\lambda_1 + r_2\lambda_2)\left[ \sum_{p,q}^{K,K} D_{ip}D_{jq}I_4 \begin{pmatrix} Q_{ii}^{(1)} & Q_{ij}^{(1)} & Q_{ip}^{(1)} & Q_{iq}^{(1)} \\ & Q_{jj}^{(1)} & Q_{jp}^{(1)} & Q_{jq}^{(1)} \\ & & Q_{pp}^{(1)} & Q_{pq}^{(1)} \\ & & & Q_{qq}^{(1)} \end{pmatrix} + \sum_{p,q}^{M,M} E_{ip}E_{jq}I_4 \begin{pmatrix} Q_{ii}^{(1)} & Q_{ij}^{(1)} & R_{ip}^{(1)} & R_{iq}^{(1)} \\ & Q_{jj}^{(1)} & R_{jp}^{(1)} & R_{jq}^{(1)} \\ & & T_{pp}^{(1)} & T_{pq}^{(1)} \\ & & & T_{qq}^{(1)} \end{pmatrix} \right. \\
&- \sum_{p,q}^{K,M} D_{ip}E_{jq}I_4 \begin{pmatrix} Q_{ii}^{(1)} & Q_{ij}^{(1)} & Q_{ip}^{(1)} & R_{iq}^{(1)} \\ & Q_{jj}^{(1)} & Q_{jp}^{(1)} & R_{jq}^{(1)} \\ & & Q_{pp}^{(1)} & R_{pq}^{(1)} \\ & & & T_{qq}^{(1)} \end{pmatrix} - \sum_{p,q}^{M,K} E_{ip}D_{jq}I_4 \begin{pmatrix} Q_{ii}^{(1)} & Q_{ij}^{(1)} & R_{ip}^{(1)} & Q_{iq}^{(1)} \\ & Q_{jj}^{(1)} & R_{jp}^{(1)} & Q_{jq}^{(1)} \\ & & T_{pp}^{(1)} & R_{pq}^{(1)} \\ & & & Q_{qq}^{(1)} \end{pmatrix} \left. \right],
\end{aligned}
$$

$$
\begin{aligned}
\frac{dQ_{ij}^{(1)}}{d\tilde{\alpha}} = \eta &\left[ \sum_{p=1}^{M} E_{ip}I_3 \begin{pmatrix} Q_{ii}^{(1)} & (\lambda_1 + \lambda_2)Q_{ij}^{(1)} - \lambda_1\lambda_2 Q_{ij}^{(0)} & R_{ip}^{(1)} \\ & * & (\lambda_1+\lambda_2)R_{jp}^{(1)} - \lambda_1\lambda_2 R_{jp}^{(0)} \\ & & T_{pp}^{(1)} \end{pmatrix} \right. \\
&- \sum_{p=1}^{K} D_{ip}I_3 \begin{pmatrix} Q_{ii}^{(1)} & (\lambda_1 + \lambda_2)Q_{ij}^{(1)} - \lambda_1\lambda_2 Q_{ij}^{(0)} & Q_{ip}^{(1)} \\ & * & (\lambda_1+\lambda_2)Q_{jp}^{(1)} - \lambda_1\lambda_2 Q_{jp}^{(0)} \\ & & Q_{pp}^{(1)} \end{pmatrix} \\
&+ \sum_{p=1}^{M} E_{jp}I_3 \begin{pmatrix} Q_{jj}^{(1)} & (\lambda_1 + \lambda_2)Q_{ji}^{(1)} - \lambda_1\lambda_2 Q_{ji}^{(0)} & R_{jp}^{(1)} \\ & * & (\lambda_1+\lambda_2)R_{ip}^{(1)} - \lambda_1\lambda_2 R_{ip}^{(0)} \\ & & T_{pp}^{(1)} \end{pmatrix} \\
&- \sum_{p=1}^{K} D_{jp}I_3 \begin{pmatrix} Q_{jj}^{(1)} & (\lambda_1 + \lambda_2)Q_{ji}^{(1)} - \lambda_1\lambda_2 Q_{ji}^{(0)} & Q_{jp}^{(1)} \\ & * & (\lambda_1+\lambda_2)Q_{ip}^{(1)} - \lambda_1\lambda_2 Q_{ip}^{(0)} \\ & & Q_{pp}^{(1)} \end{pmatrix} \left. \right] \\
&+ \eta^2(r_1\lambda_1^2 + r_2\lambda_2^2)\left[ \sum_{p,q}^{K,K} D_{ip}D_{jq}I_4 \begin{pmatrix} Q_{ii}^{(1)} & Q_{ij}^{(1)} & Q_{ip}^{(1)} & Q_{iq}^{(1)} \\ & Q_{jj}^{(1)} & Q_{jp}^{(1)} & Q_{jq}^{(1)} \\ & & Q_{pp}^{(1)} & Q_{pq}^{(1)} \\ & & & Q_{qq}^{(1)} \end{pmatrix} + \sum_{p,q}^{M,M} E_{ip}E_{jq}I_4 \begin{pmatrix} Q_{ii}^{(1)} & Q_{ij}^{(1)} & R_{ip}^{(1)} & R_{iq}^{(1)} \\ & Q_{jj}^{(1)} & R_{jp}^{(1)} & R_{jq}^{(1)} \\ & & T_{pp}^{(1)} & T_{pq}^{(1)} \\ & & & T_{qq}^{(1)} \end{pmatrix} \right. \\
&- \sum_{p,q}^{K,M} D_{ip}E_{jq}I_4 \begin{pmatrix} Q_{ii}^{(1)} & Q_{ij}^{(1)} & Q_{ip}^{(1)} & R_{iq}^{(1)} \\ & Q_{jj}^{(1)} & Q_{jp}^{(1)} & R_{jq}^{(1)} \\ & & Q_{pp}^{(1)} & R_{pq}^{(1)} \\ & & & T_{qq}^{(1)} \end{pmatrix} - \sum_{p,q}^{M,K} E_{ip}D_{jq}I_4 \begin{pmatrix} Q_{ii}^{(1)} & Q_{ij}^{(1)} & R_{ip}^{(1)} & Q_{iq}^{(1)} \\ & Q_{jj}^{(1)} & R_{jp}^{(1)} & Q_{jq}^{(1)} \\ & & T_{pp}^{(1)} & R_{pq}^{(1)} \\ & & & Q_{qq}^{(1)} \end{pmatrix} \left. \right],
\end{aligned}
$$

$$
\begin{aligned}
\frac{dR_{in}^{(0)}}{d\tilde{\alpha}} = \eta &\left[ \sum_{p=1}^{M} E_{ip}I_3 \begin{pmatrix} Q_{ii}^{(1)} & R_{in}^{(1)} & R_{ip}^{(1)} \\ & * & T_{np}^{(1)} \\ & & T_{pp}^{(1)} \end{pmatrix} - \sum_{p=1}^{K} D_{ip}I_3 \begin{pmatrix} Q_{ii}^{(1)} & R_{in}^{(1)} & Q_{ip}^{(1)} \\ & * & R_{pn}^{(1)} \\ & & Q_{pp}^{(1)} \end{pmatrix} \right],
\end{aligned}
$$

$$
\begin{aligned}
\frac{dR_{in}^{(1)}}{d\tilde{\alpha}} = \eta &\left[ \sum_{p=1}^{M} E_{ip}I_3 \begin{pmatrix} Q_{ii}^{(1)} & (\lambda_1 + \lambda_2)R_{in}^{(1)} - \lambda_1\lambda_2 R_{in}^{(0)} & R_{ip}^{(1)} \\ & * & (\lambda_1+\lambda_2)T_{np}^{(1)} - \lambda_1\lambda_2 T_{np}^{(0)} \\ & & T_{pp}^{(1)} \end{pmatrix} \right. \\
&- \sum_{p=1}^{K} D_{ip}I_3 \begin{pmatrix} Q_{ii}^{(1)} & (\lambda_1 + \lambda_2)R_{in}^{(1)} - \lambda_1\lambda_2 R_{in}^{(0)} & Q_{ip}^{(1)} \\ & * & (\lambda_1+\lambda_2)R_{pn}^{(1)} - \lambda_1\lambda_2 R_{pn}^{(0)} \\ & & Q_{pp}^{(1)} \end{pmatrix} \left. \right]
\end{aligned}
\tag{4}
$$

, and

$$\frac{dD_{ij}}{d\tilde{\alpha}} = \eta \left[ \sum_{p=1}^{M} E_{ip} I_2 \begin{pmatrix} Q_{jj}^{(1)} & R_{jp}^{(1)} \\ & T_{pp}^{(1)} \end{pmatrix} - \sum_{p=1}^{K} D_{ip} I_2 \begin{pmatrix} Q_{jj}^{(1)} & Q_{jp}^{(1)} \\ & Q_{pp}^{(1)} \end{pmatrix} \right.$$
$$\left. + \sum_{p=1}^{M} E_{jp} I_2 \begin{pmatrix} Q_{ii}^{(1)} & R_{ip}^{(1)} \\ & T_{pp}^{(1)} \end{pmatrix} - \sum_{p=1}^{K} D_{jp} I_2 \begin{pmatrix} Q_{ii}^{(1)} & Q_{ip}^{(1)} \\ & Q_{pp}^{(1)} \end{pmatrix} \right], \qquad (5)$$
$$\frac{dE_{in}}{d\tilde{\alpha}} = \eta \left[ \sum_{p=1}^{M} F_{pn} I_2 \begin{pmatrix} Q_{ii}^{(1)} & R_{ip}^{(1)} \\ & T_{pp}^{(1)} \end{pmatrix} - \sum_{p=1}^{K} E_{pn} I_2 \begin{pmatrix} Q_{jj}^{(1)} & R_{jp}^{(1)} \\ & T_{pp}^{(1)} \end{pmatrix} \right]$$

.

## A.3  Dependence of learning trajectory on initial conditions on macroscopic parameters

Figure A.1:   Dynamics of generalization error $\varepsilon_g$ and order parameters $Q_{ij}$ and $R_{in}$ computed with macroscopic system, and its variability by random weight initialization. Network size: $N$-2-1. Learning rate: $\eta = 0.1$. Eigenvalues of $\Sigma$: $\lambda_1 = 0.3$ with multiplicity $0.5N$, $\lambda_2 = 1.7$ with multiplicity $0.5N$. Black lines: dynamics of $\varepsilon_g$. Blue lines: $Q_{11}, Q_{12}, Q_{22}$. Green lines: $R_{11}, R_{12}, R_{21}, R_{22}$. (a) $N = 10^5$, (b) $N = 10^7$. In both figures, solid curves and shades represent mean and standard deviation of 100 trials, respectively (note that mean and standard deviation of loss are computed in logarithmic scale).

In the statistical mechanical formulation, by considering $N$ as large, the dynamics of the system is reduced to macroscopic differential equations with small ($N$-independent) dimensions. The macroscopic system we derived is deterministic in the sense that randomness brought by stochastic gradient descent is vanished. However, note that the trajectory of the macroscopic state can vary in accordance with its initial condition. Figure A.1 shows this variability with shades.

How does the initial condition affect the learning trajectory? Consider a typical initialization that the microscopic parameters $\boldsymbol{J}_1, \boldsymbol{J}_2, \boldsymbol{B}_1$ and $\boldsymbol{B}_2$ are initialized as $(\boldsymbol{J}_i)_k, (\boldsymbol{B}_n)_k \overset{\text{i.i.d.}}{\sim} \mathcal{N}(0, 1/N)$. Then the mean and variance of corresponding initial macroscopic parameters $Q$, $R$ and $T$ are

$$\mathbb{E}[Q_{ii}^{(e)}] = \mu_e, \quad \mathbb{V}[Q_{ii}^{(e)}] = \frac{3\mu_{2e}}{N}, \quad \mathbb{E}[Q_{ij}^{(e)}] = 0, \quad \mathbb{V}[Q_{ij}^{(e)}] = \frac{\mu_{2e}}{N},$$
$$\mathbb{E}[R_{in}^{(e)}] = 0, \quad \mathbb{V}[R_{in}^{(e)}] = \frac{\mu_{2e}}{N},$$
$$\mathbb{E}[T_{nn}^{(e)}] = \mu_e, \quad \mathbb{V}[T_{nn}^{(e)}] = \frac{3\mu_{2e}}{N}, \quad \mathbb{E}[T_{nm}^{(e)}] = 0, \quad \mathbb{V}[T_{nm}^{(e)}] = \frac{\mu_{2e}}{N}$$

With $N \to \infty$, these probabilistic parameters converge to $(Q^{(e)}, R^{(e)}, T^{(e)}) = (\mu_e I_K, 0, \mu_e I_M)$. However, the solution trajectory starting from just $(\mu_e I_K, 0, \mu_e I_M)$ cannot break the weight symmetry at all. To argue practical learning trajectory, we have to consider the initial value slightly off from that point. How close the initial condition is to that point affects how long it takes to break the weight

symmetry, that is, the plateau length. This is why Figure A.1 (b) with $N = 10^7$ exhibits plateau slightly longer than that of Figure A.1 (a) with $N = 10^5$.