[Reviews · NeurIPS 2019]

Reviewer 1



- Their analysis of the dependence of the plateau on the spectral density is based on an unrealistic spectrum where the eigenvalues are highly degenerate. It would make more sense to show results for data with low-dimensional structure, in which the first one or two are non-zero, and the rest are either zero or epsilon small. Do the conclusions for the two eigenvalues case still hold in this example? It is hard for me to see what I should learn from figures 5 and 6. - The dependence of the learning dynamics on the spectral properties of the input data is not new and was previously studies by Saxe et al. (ArXiv, 2013) for simple linear networks. It would be appropriate if these results were mentioned or discussed in the text. - In their analysis, the authors do not consider the importance of initial conditions of the weights. It has been previously showed that the initial conditions have a big impact on the trainability and learning dynamics of these networks. In this case, they would be defined as the initial conditions on the order parameters Q, R, and D. - The analysis here seems tractable only for networks with a small number of hidden units. This regime is different from many recent studies of mean-field, that assume the number of hidden units is large at the thermodynamic limit. It is not clear to me if the phenomena of the plateaus, in this case, is the result of the bottleneck structure where a very high dimensional input (N) is projected to an O(1) number of non-linear units. Are the plateaus a result of a bottleneck in the network? If we were to look at a deep network, where the number of hidden units is also at the thermodynamic limit (Schoenholtz, Glimer, Ganguly & Sohl-Dickstein, 2016), will the plateaus arise from the final layer in this case? - Some of the mean-field derivations are cumbersome and unclear. For example, it is not clear to me why the authors needed to define the variable Omega and its dynamics. It does not seem to serve any conclusion, and t was hard to understand. For example, define more functions, f,g,h without stating what they are. Furthermore, the use of g, which was used above for the transfer function is confusing. ***update *** After the authors' response, I am still not convinced about the meaningfulness of the simple spectrum. Furthermore, the comments I had, as the initial conditions and the width of the hidden layer were not addressed properly. However, I agree with reviewer #2 that the work is interesting, original, and should be published. I am recommending to publish this work but strongly urge the authors to address some of my concerns. First and foremost, the clarity and readability of the paper should be improved to make it accessible to a wider audience.

Reviewer 2



I think this is a very nice and original contribution trying to bridge properties of simple models of neural networks, the teacher-student soft committee machine in the present case, with empirical observations available in deep learning. Adjustment of the models to match the observed behaviour seems to be a very valuable way to proceed towards better understanding of the deep learning phenomena. The paper is well written. Comments and questions: ** Can the authors define the relation between number of epochs and number normalized steps? In particular in Fig. 1, did the system see only 10*Epoch samples or was it the whole MNIST passed trough several times by the SGD. ** To give a better idea of how good is the neural network the authors are using, can they state the accuracy on the test set corresponding to Fig. 1? ** The authors summarize one of their contribution by saying: "By analyzing the macroscopic system we derived, we showed that the dynamics of learning depends only on the eigenvalue distribution of the covariance matrix of the input data, provided that the learning rate is sufficiently small." This should be stated more precisely. Surely the dynamics of learning depends also on the way the labels were generated, which is not considered in this sentence. ** The plateau phenomena is intimately related to the specialization of hidden student units ot the teacher units, I think it would be valuable if the authors discuss this connection quantitatively and evaluate their theory in this respect. In particular the authors conclude "Considering this, the claim that the plateau phenomenon does not occur in learning of MNIST is controversy; this suggests the possibility that we are observing quite long and low plateaus." Shouldn't the specialization of hidden units or the lack of there-off be a good measure to resolve this "controversy"? ------ post-feedback I have read the other reviews and the author's feedback. I maintain my score. The problem this paper addresses is in my opinion important. At the same time I urge the authors to consider all comments from the reviews to make their paper clearer to the NeurIPS audience.

Reviewer 3



I read the paper, but did not check the mathematical details. I think the paper is interesting and it seems that it relies on standard mathematical analysis. I think the results are important and advance the knowledge about neural nets.

[Author Response · NeurIPS 2019]

Thank you all for helpful reviewing. We clarify our viewpoint with respect to some of your questions.

**Is the assumption about data spectrum unrealistic?** The reviewer 1 pointed out that our analysis is based on "an unrealistic spectrum where the eigenvalues are highly degenerative". However, we think there may be some misunderstanding. Our formulation is not limited to some specific eigenvalue distributions. The macroscopic equations (3) and (5) we derived have no limitation about the number of distinct eigenvalues. Besides, just because there are one or two distinct eigenvalues (as considered in section 4), does not mean that the data distributes within $\leq 2$ dimensional subspace. For instance, if there are two distinct eigenvalues, $\lambda_1(> 0)$ of multiplicity $N_1$ and $\lambda_2(> 0)$ of multiplicity $N_2$, the data span $N_1 + N_2$ dimensional subspace. Note that $N_1 + N_2$ can be 2, the case with very degenerated input distribution, and it can be $N$, the case with non-degenerated input distribution.

Though it is unusual that $N_i$ eigenvalues strictly coincide in real dataset, we think that it is meaningful to simplify the eigenvalue distribution to a simple one that has controllable first and second moments, for the main purpose of understanding the relationship between macroscopic data statistics and learning dynamics in an interpretable way.

**Difference from recent works on learning dynamics.** As the reviewer 1 pointed out, some recent works including Saxe et al. [2019] analyze the relationship between learning dynamics and data structure (spectrum) with linear networks. However, linear networks do not exhibit the plateau phenomenon which is our main interest, since there is no need to occur specialization of weights in linear nets (see Aubin et al. [2018], for example). To our best knowledge, it is the first time for the plateau phenomenon to be discussed in relation to the statistical nature of the data. We are willing to add a mention about this in our camera-ready manuscript if this work is accepted.

**On initial conditions of weights.** The importance of initial conditions on successful learning has been shown in several works, including Schoenholz et al. [2016]; they suggests that if the initial condition is close to the edge of chaos, the depth scale of the network will become larger and the network will becomes more trainable even if it consists of dozens of layers. However, we think that this effect is limited in our situation, because what we focus on is shallow networks which consist of only two layers.

**Meaning of order parameter $\Omega$.** We introduced $\Omega^{(e)}$ as grouping the $e$-th order parameters of the first layers as $\Omega^{(e)} := (Q^{(e)}, R^{(e)}, T^{(e)})$, rather than as a new order parameter. Likewise, we defined $\chi$ in order to group together the order parameters of the second layers. The equation shown above p.6 containing $f^{(e)}$, $g$ and $h$ is for emphasizing the dependency between order parameters (the specific form of $f^{(e)}$, $g$ and $h$ is written in the equations (3) and (5)); the important thing is that the 'speed' of $e$-th order parameters of first layers depends *only* on 1-st and $(e+1)$-th order parameters of first layers, $\Omega^{(1)}$ and $\Omega^{(e+1)}$ (and order parameters of second layers $\chi$). Together with the equation $\Omega^{(d)} = \cdots$ in p.6, which indicates that $d$-th order parameters can be represented by $(< d)$-th order parameters, we can obtain the closed system of differential equations of order parameters which consist of $\Omega^{(0)}, \Omega^{(1)}, \ldots, \Omega^{(d-1)}$ and $\chi$.

**Contribution and takeaway of our works.** Plateau phenomenon has been researched mainly with regards to the structure of neural networks, and the relationship with the data has been overlooked (Saad and Solla [1995], Amari et al. [2018]). However, our work focuses on the statistics of the data learned and its impact on the learning dynamics, including plateau phenomenon. The statistical mechanical method we used is a powerful tool for analyzing learning dynamics of nonlinear neural networks. Our main contribution is to extend this method to generalized cases where the input data has arbitrary covariance and to suggest the framework for understanding the relationship between data statistics and plateau phenomenon. The takeaway is the following: the plateau phenomenon specific to learning dynamics of nonlinear neural network is heavily dependent to the data learned. By deriving the learning dynamics with generalized data statistics, we can develop the theory of plateau phenomenon in more realistic settings.

# References

S. Amari, T. Ozeki, R. Karakida, Y. Yoshida, and M. Okada. *Neural computation*, 30(1):1–33, 2018.

B. Aubin, A. Maillard, F. Krzakala, N. Macris, L. Zdeborová, et al. In *NeurIPS*, pages 3223–3234, 2018.

D. Saad and S. A. Solla. *Phys. Rev. E*, 52(4):4225, 1995.

A. M. Saxe, J. L. McClelland, and S. Ganguli. *PNAS*, 116(23):11537–11546, 2019.

S. S. Schoenholz, J. Gilmer, S. Ganguli, and J. Sohl-Dickstein. *arXiv preprint arXiv:1611.01232*, 2016.


[Meta-Review · NeurIPS 2019]

This paper provides an analysis on dynamics of online learning of two-layer neural networks under the teacher-student scenario. The analysis extends that by Saad and Solla (1995) by considering a covariance matrix of the input which may not be proportional to the identity matrix. The main contribution of this paper is the finding that the plateau phenomenon observed in learning dynamics of nonlinear neural networks depends on statistics of input data. The three reviewers rated this paper above the acceptance threshold, mentioning originality and importance of the contribution of this paper. At the same time, two reviewers raised concern about clarity of presentation. The clarity concern still remains after the authors' rebuttal and subsequent discussion among the reviewers. I would recommend acceptance of this paper in view of its originality and potential importance. I would like to strongly encourage the authors to take into account the review comments seriously to improve clarity of their presentation. I would also like to supplement some of the review comments in the following: - On initial condition of weights: As Reviewer #1 mentioned, the initial condition of weights in the numerical experiments in Section 4 should be specified explicitly in view of reproducibility. In the rebuttal the authors simply claimed their belief that the effect of initial conditions should be limited, but showing numerical results with several different initial conditions would provide a direct and explicit evidence. - On number of hidden units: In this study the numbers of hidden units are K and M for the student and teacher networks, respectively. Then the total number of order parameters are roughly O(d(K+M)^2). As Reviewer #1 mentioned, on the other hand, there have been several papers studying macroscopic description of deep neural networks with random weights, where the limit of the numbers of units in hidden layers to infinity is typically considered. A natural question, which is actually one of the comments by Reviewer #1, is whether the plateau phenomenon still persists in the latter limit as well. It would be nice if some comments toward such extensions will be given. - On specialization of student hidden units: I feel that little insights have been provided as to why the plateau phenomenon would happen depending on the input covariance. As commented by Reviewer #2, specialization of student hidden units, or equivalently, as the authors briefly mention in lines 20-24, breaking of the intrinsic symmetry with respect to exchange of student hidden units will be responsible for the plateau phenomenon. The authors nevertheless did not discuss this issue any further, even in their rebuttal. It is important to investigate the mechanism causing the plateau phenomenon rather than just to demonstrate that the non-identity covariance could cause plateaux. - On reference list: In References, currently only 8 papers are listed, which are very small in number compared with typical NeurIPS submissions. In fact, in the rebuttal the authors cited 4 more papers in their additional discussion, and I suspect that there would be even more regarding the above mentioned points. These papers should be included in the reference list, as the references exempted from the page limit. Minor points: - Line 74: N in 1 \le n \le N should read M. In the displayed equation that follows the line, y_N should be y_M.